# SAR Imaging Algorithm of Ocean Waves Based on Optimum Subaperture

**DOI:** 10.3390/s22031299

**Published:** 2022-02-08

**Authors:** Yawei Zhao, Xianen Wei, Jinsong Chong, Lijie Diao

**Affiliations:** 1National Key Lab of Microwave Imaging Technology, Beijing 100190, China; zhaoyawei17@mails.ucas.ac.cn (Y.Z.); weixianen20@mails.ucas.ac.cn (X.W.); diaolijie21@mails.ucas.ac.cn (L.D.); 2Aerospace Information Research Institute, Chinese Academy of Sciences, Beijing 100190, China; 3School of Electronics, Electrical and Communication Engineering, University of Chinese Academy of Sciences, Beijing 100049, China

**Keywords:** SAR, ocean waves, optimum subaperture, refocusing

## Abstract

Synthetic Aperture Radar (SAR) is widely applied to the field of ocean remote sensing. Clear SAR images are the basis for ocean information acquisitions, such as parameter retrieval of ocean waves and wind field inversion of the ocean surface. However, the SAR ocean images are usually blurred, which seriously affects the acquisition of ocean information. The reasons for the wave blurring in SAR images mainly include the following two aspects. One is that when SAR observes the ocean, the motion of ocean waves will have a greater impact on imaging quality. The other is that the ocean’s surface is seriously decorrelated within the integration time. In order to obtain clear SAR images of ocean waves, a SAR imaging algorithm of ocean waves based on the optimum subaperture is proposed, aiming at the above two aspects. The optimum focus setting of the ocean waves is calculated, drawing support from the azimuth phase velocity of the dominant wave. The optimum subaperture is further calculated according to the proposed new evaluation, namely, *F*. Finally, according to the optimum focus setting and the optimum subaperture, the dominant wave is refocused, and a clear SAR image of the dominant wave can be obtained. The proposed algorithm was applied to airborne L-band and P-band SAR data. Furthermore, the proposed algorithm was compared with present methods, and the results sufficiently demonstrated the effectiveness and superiority of the proposed algorithm.

## 1. Introduction

Synthetic aperture radar (SAR) is widely applied to the field of ocean remote sensing with the obvious advantages of all-day, all-weather, high resolution and wide swath coverage. To obtain a higher azimuth resolution, it is necessary to increase the SAR integration time to achieve signal accumulation. However, the random motion of ocean waves will have a greater impact on SAR image quality when observing the ocean. If the motion of ocean waves is not taken into account, the ocean waves in the SAR image will be extremely blurred or even invisible [1,2]. Not only will this restrict the imaging ability of SAR for some small-scale ocean waves [3], but also it has a serious influence on the subsequent parameter retrieval of ocean waves [4,5,6] and wind field inversion of the ocean surface [7,8]. Therefore, it is essential to improve the SAR image quality of ocean waves by means of some methods.

The present methods of SAR image quality improvement can be divided into two major categories. The first type is image processing methods, which are mainly based on noise suppression, such as Lee filtering [9], Kuan filtering [10] and so on. In recent years, machine learning methods have also been applied to the study of SAR image denoising, and many representative methods have been proposed [11,12,13,14,15]. But image denoising cannot solve the problem of wave blurring caused by imaging. The second is signal processing methods, which are usually implemented by improving imaging algorithms. The imaging algorithm based on focus setting is a common algorithm for ocean wave imaging. The focus setting is defined as the difference between azimuth matched filter speed and platform speed, and the focus setting that makes the target best-focused is called the optimum focus setting [16,17]. Regarding the studies on SAR focus imaging for ocean waves, most scientists have demonstrated that surface waves could achieve the optimum focus when the focus setting is equal to half of its azimuth phase speed [16,17,18,19,20,21,22]. However, some scientists have confirmed that the optimum focus setting of ocean waves has a clear deviation by half of its azimuth phase speed under specified conditions through simulation and experimental data processing [23,24]. For example, the condition that the SAR integration time is short or that the wave propagation direction is large relative to the azimuth. To solve the problem that the optimum focus setting of ocean waves is not necessarily half of its azimuth phase velocity, Wei [25] proposed an airborne SAR imaging algorithm for ocean waves based on optimum focus setting. However, the above studies did not consider the impact of ocean surface coherence time on the quality of ocean wave images.

A large number of simulated and experimental results show that the coherence time of the ocean’s surface is generally a few milliseconds to hundreds of milliseconds [26,27], while the SAR integration time is a few seconds to tens of seconds. The SAR integration time is considerably longer than the coherence time of the ocean surface. Owing to the random motion of the ocean surface during SAR integration time, serious decorrelation of the scattering units will occur. Severe decorrelation of the scattering units will lead to a serious decrease in SAR echo coherence, which will affect the focusing effect of ocean waves.

In order to ensure the coherence of SAR echo when ocean wave imaging, the subaperture technology is considered in this paper. At present, some important aspects of SAR imaging technology have introduced the idea of subaperture processing, such as subaperture real-time imaging [28] and SAR subaperture imaging for high squint [29,30]. The subaperture imaging can reduce the integration time of SAR images. So, the subaperture technology can ensure the coherence of echo.

Therefore, the reasons for blurred SAR wave images include the following two aspects. One is that the motion of the ocean waves causes the waves to be blurred. The other is that the severe decorrelation of the ocean’s surface during the integration time affects the imaging quality of ocean waves. For the above two aspects, a SAR imaging algorithm of ocean waves based on the optimum subaperture is proposed in this paper. In order to calculate the optimum subaperture, a new evaluation *F* is calculated, designating for SAR single-look complex (SLC) data. Then, optimum focus imaging of ocean waves is achieved by combining the optimum focus setting. The proposed algorithm was applied to airborne L-band and P-band SAR data. Then, the processing results of the proposed algorithm were compared with the original SAR image, the refocusing SAR image based on half of azimuth phase speed [16,17,18,19,20,21,22] and the refocusing SAR image based on the optimum focus setting [25]. At the same time, quantitative analysis was carried out by calculating the contrast of SAR images, the relative normalized modulation of image gray and the modulation difference of positive and negative gray, which sufficiently demonstrated the effectiveness and superiority of the proposed algorithm. Finally, the applicability of this algorithm is analyzed.

The rest of the paper is organized as follows. In Section 2, the proposed algorithm is introduced in detail. In Section 3, the results of airborne L-band and P-band field data are given. In Section 4, the proposed algorithm is compared with the original SAR image, the refocusing SAR image based on half of azimuth phase speed and the refocusing SAR image based on the optimum focus setting, which demonstrates the effectiveness and superiority of the proposed algorithm. The discussion is given in Section 5. Finally, conclusions are made in Section 6.

## 2. SAR Imaging Algorithm of Ocean Waves Based on Optimum Subaperture

In this section, a SAR imaging algorithm of ocean waves based on the optimum subaperture is proposed, which is designated for single-look complex (SLC) data. The flow chart of the proposed algorithm is shown in Figure 1. As can be seen, the algorithm can be divided into five parts: the selection of sub-block data, the calculation of the focus setting variation section, the calculation of the optimum focus setting, the calculation of the optimum subaperture and the refocusing of the panoramic image. The above five parts will be detailed below.

### 2.1. Selection of Sub-Block Data

To calculate the optimum focus setting and optimum subaperture, the sub-block data is chosen from the SAR SLC data. The main purpose of choosing the sub-block is to reduce the interference of artificial targets on the ocean’s surface to the calculation results. Therefore, the selection of the sub-block is according to the following two constraints: (1) this sub-block needs to contain ocean waves and (2) does not contain interference information other than ocean waves.

After choosing the sub-block data, two operations have to be performed. On the one hand, the sub-block amplitude data is calculated for the calculation of the focus setting variation section. On the other hand, a fast Fourier transform (FFT) along the azimuth direction is performed to obtain the range-Doppler domain data, which is used for the optimum focus setting calculation and the optimum subaperture calculation.

### 2.2. Calculation of Focus Setting Variation Section

After obtaining the amplitude of the sub-block data, it is utilized to calculate the focus setting variation section Δ*V*. The sub-block amplitude data needs to be calibrated in the first instance, which includes slant-to-ground conversion, multi-look processing and range energy normalization [31]. After the above calibrations, the sub-block image is transformed into the wavenumber domain by two-dimensional FFT. Then, the wavenumber vector $k_{s}$ of the dominant wave is obtained from the two-dimensional wavenumber spectrum, which is expressed as
(1)$$k_{s}=\left[\begin{array}{c} k_{rs}\\ k_{as} \end{array}\right]$$
where $k_{rs}$ and $k_{as}$ are the range wavenumber and the azimuth wavenumber of the dominant wave in the SAR image, respectively.

The range wavenumber $k_{ro}$ of the dominant wave on the real ocean surface is equal to the $k_{rs}$ of the dominant wave in the SAR image. However, affected by scanning distortion, the azimuth wavenumber $k_{ao}$ of the dominant wave on the real ocean surface is typically different from the of the dominant wave in the SAR image [32,33]. The relationship between $k_{ao}$ and $k_{as}$ can be expressed as
(2)$$k_{ao}=k_{as}+\frac{\sqrt{gk_{o}}}{V}$$
where g is the acceleration of gravity, $k_{o}=\sqrt{k_{ao}^{2}+k_{ro}^{2}}$ is the wavenumber of the dominant wave on the real ocean surface and V is the platform speed. By solving Equation (2), the azimuth wavenumber $k_{ao}$ of the dominant wave on the real ocean surface can be obtained. Afterwards, the angle ϕ between the dominant wave propagation direction and the azimuth direction can be obtained as
(3)$$\phi =\mathrm{arc}\tan\frac{k_{ro}}{k_{ao}}$$

After calculating the propagation direction of the dominant wave, the azimuth phase velocity $C_{a}$ can be calculated as
(4)$$C_{a}=\frac{\sqrt{g/k_{o}}}{\cos\phi }$$

However, due to the complexity of ocean wave motion, the optimum focus setting is not necessarily $C_{a}/2$ in many practical cases. Therefore, after obtaining the azimuth phase velocity $C_{a}$ of the dominant wave, the next step is to set a focus setting variation section ΔV. The ΔV is given by
(5)$$\Delta V\in \left[\frac{C_{a}}{2}-k\Delta v,\frac{C_{a}}{2}+k\Delta v\right]$$

Referring to the literature [25] for the values of *k* and Δ*v*, they can generally take *k* = 8 and Δ*v =* 1 *m/s*.

### 2.3. Calculation of the Optimum Focus Setting

After obtaining the focus setting variation section ΔV, the sub-block image is refocused using each velocity $\Delta V_{i}\left(i=1,2,\ldots ,2k+1\right)$ to calculate the optimum focus setting. The range-Doppler domain data obtained in Section 2.1 is required as the input of the next process.

In this process, the range-Doppler domain data is multiplied by the complex conjugate of the azimuth-matched filter used in the imaging process, and the uncompressed data in the range-Doppler domain are obtained. Afterwards, according to different focus settings $\Delta V_{i}$, the speed $W_{i}$ of the new azimuth-matched filter is calculated by:(6)$$W_{i}=V-\Delta V_{i}$$

Subsequently, a new azimuth matched filter function $h_{i}\left(\tau \right)$ is generated based on the speed $W_{i}$
(7)$$h_{i}\left(\tau \right)=\exp\left(j\frac{2\pi W_{i}^{2}\tau^{2}}{\lambda R_{0}}\right)$$
where τ represents the azimuth time, λ represents the radar wavelength and $R_{0}$ represents the slant range. To obtain the refocused images, azimuth compression is performed for the uncompressed data by means of $h_{i}\left(\tau \right)$. Furthermore, the refocused image is calibrated. The calibrations include slant-to-ground conversion, multi-look processing and range energy normalization.

Next, the calibrated SAR images are compared to determine the optimum focus setting $\Delta V_{opt}$. Therefore, the SAR image is transformed into a two-dimensional wavenumber domain through a two-dimensional FFT. Then, the peak-to-background ratio (*PBR*) of the dominant wave energy in the wavenumber spectrum is calculated. The definition of *PBR* is as follows [18,34]:(8)$$PBR=\frac{{\left(S_{I}\right)}_{\max}}{\langle S_{n}\rangle }$$
where ${\left(S_{I}\right)}_{\max}$ and $\langle S_{n}\rangle$ represent the peak value and noise floor of the SAR image spectrum, respectively.

Finally, the maximum of PBR
is calculated by:(9)$$PBR_{\max}\left(\Delta V_{i}\right)=\underset{\Delta V_{i}}{\max}\left(PBR_{i}\left(\Delta V_{i}\right)\right),i=1,2,\ldots ,2k+1$$
where $PBR\left(\Delta V_{i}\right)$ represents the PBR
changing with $\Delta V_{i}$. The focus setting corresponding to $PBR_{\max}\left(\Delta V_{i}\right)$ is the optimum focus setting $\Delta V_{opt}$ for the dominant wave.

### 2.4. Calculation of Optimum Subaperture

The calculation of the optimum subaperture will be introduced in this subsection. After the optimum focus setting $\Delta V_{opt}$ is obtained, the calculation of the optimum subaperture needs to be carried out with the help of $\Delta V_{opt}$. The division method of the Doppler spectrum is used to perform subaperture segmentation [28,35]. In this process, subaperture segmentation is performed on the sub-block range-Doppler domain data, where the subaperture bandwidth selected for each segmentation is:(10)$$Ba_{i}=\frac{i}{N}\cdot Ba,\;\left(i=1,2,\ldots ,N\right)$$
where Ba represents the full-aperture bandwidth of Doppler.

By calculating the focusing effect of the real ocean surface data under different subapertures, the curve $PBR\left(Ba_{i}\right)$, which changes with subaperture bandwidth, is obtained as shown in Figure 2a, where $PBR\left(Ba_{i}\right)$ is the change in PBR with $Ba_{i}$. From Figure 2a, it can be found that with the increase in subaperture bandwidth, the refocusing effect of the dominant wave becomes worse. At the same time, the curve of the equivalent number of looks (*ENL*) changing with the subaperture bandwidth is shown in Figure 2b. The *ENL* is calculated by
(11)$$ENL=\frac{\mu^{2}}{\sigma^{2}}$$
where μ and $\sigma^{2}$ represent the mean and the variance of the image, respectively. From Figure 2b, it can be found that with an increasing subaperture bandwidth, the signal-to-noise ratio of the SAR image gradually improves.

By comparing Figure 2a,b, it can be found that the use of subaperture segmentation technology to improve the refocusing effect of dominant waves will lead to a decrease in the signal-to-noise ratio of the SAR image. Therefore, to obtain the optimum subaperture, this paper proposed a new evaluation, namely, F, referring to the calculation method of the *F*-measure [36,37] in machine learning, which considers the peak-to-background ratio and the equivalent number of looks. The F can be calculated by:(12)$$F=\frac{P\times E}{P+E}$$
among them,
(13)$$P=\frac{PBR\left(Ba_{i}\right)-PBR_{\min}\left(Ba_{i}\right)}{PBR_{\max}\left(Ba_{i}\right)-PBR_{\min}\left(Ba_{i}\right)}$$
(14)$$E=\frac{ENL-ENL_{\min}}{ENL_{\max}-ENL_{\min}}$$

The variation curves of P,E,F with the subaperture bandwidth obtained according to real ocean surface data are shown in Figure 3. The red dotted line is the change in P with subaperture bandwidth. The green dashed line is the change in E with subaperture bandwidth. The blue solid line is the change in F with the subaperture bandwidth, in which P and E are considered at the same time. According to the curves shown in Figure 3, the new parameter F can comprehensively consider the values of *PBR* and *ENL* to realize the selection of the optimum subaperture.

F are calculated corresponding to the different subaperture bandwidths according to Equation (12), and further, the maximum of
F is obtained (15)$$F_{\max}=\underset{i}{\max}\left(F_{i}\right),\;i=1,2,\ldots ,N$$

The $i_{opt}$ corresponding to $F_{\max}$ is used to calculate the optimum subaperture, and the optimum subaperture bandwidth is expressed as
(16)$$Ba_{opt}=\frac{i_{opt}}{N}\cdot Ba$$

### 2.5. Refocusing of Panoramic Image

After obtaining the optimum focus setting $\Delta V_{opt}$ and optimum subaperture $Ba_{opt}$ of the dominant wave, the panoramic SLC data is refocused by means of $\Delta V_{opt}$ and $Ba_{opt}$. A well-focused SAR image for the dominant wave can be obtained.

## 3. Applying the Proposed Algorithm

In this section, the proposed algorithm will be applied to the field SAR data of the airborne L-band and P-band. The experimental data come from the sea trial experiment conducted by the Aerospace Information Research Institute, Chinese Academy of Sciences covering the South China Sea. The L-band and P-band data were collected on 13 September 2014 and 11 October 2014, respectively. The experimental place is located approximately 100 km south of Sanya, China. The geographic location of the SAR data on Google Earth is shown in Figure 4, where the location of the L-band data is marked in pink and that of P-band is marked in green. The parameters of the radar system are shown in Table 1. In comparison, the SAR integration time of airborne P-band SAR data is much longer than that of the L-band SAR data. Therefore, the motion and decorrelation of the ocean’s surface have a greater impact on P-band data.

The ocean wave data of different radar bands are selected for processing. The environment parameters and wave parameters are shown in Table 2. The mean wave direction is the angle between the dominant wave propagation direction and the azimuth. The parameters are derived from the European Centre for Medium-Range Weather Forecast (ECMWF) data. The two pieces of data are representative to a certain extent. On the one hand, they are different bands. On the other hand, the integration times of the two data are quite different.

The SAR image of the airborne L-band is given in Figure 5, and that of the airborne P-band is shown in Figure 6. The ocean surfaces of the two datasets are relatively uniform. As shown in Figure 5, ocean waves can be seen in some areas of the L-band SAR image. However, owing to the influences of the ocean’s surface motion and the decorrelation of the ocean surface during the integration time, the SAR image is blurry. The effect of the ocean’s surface motion on P-band data with a long SAR integration time is more obvious than that on the L-band data, in which the texture of ocean waves is scarcely invisible in Figure 6.

### 3.1. Results of Airborne L-Band SAR Data Processing

The airborne L-band SAR data shown in Figure 5 are processed through the algorithm shown in Figure 1. As shown by the yellow rectangular area in the bottom-right corner, a piece of sub-block data is selected. The sub-block data do not contain other information apart from surface waves. The obtained F changing with the subaperture bandwidth are shown in Figure 7.

It can be seen from Figure 7 that the peak of the curve is at *Ba_i_* = 37.13 Hz. Consequently, the optimum subaperture of the dominant wave is *Ba_opt_* = 37.13 Hz. That is, when the subaperture bandwidth is 9/16 of the full-aperture bandwidth, the focusing effect of the dominant wave is optimum. The panoramic data is processed by the proposed algorithm, and the refocused result is shown in Figure 8. Comparing Figure 5 and Figure 8, it can be found that the focus of the dominant wave is effective by the proposed algorithm.

### 3.2. Results of Airborne P-Band SAR Data Processing

The process shown in Figure 1 is also applied to the airborne P-band data shown in Figure 6. As shown by the yellow rectangular area in the middle-left corner, a piece of sub-block data is selected. The sub-block data is very uniform. The obtained F changes with the subaperture bandwidth are shown in Figure 9.

It can be found from Figure 9 that the peak of the curve is at *Ba_i_* = 50.63 Hz. Consequently, the optimum subaperture of the dominant wave is *Ba_opt_* = 50.63 Hz. That is, when the subaperture bandwidth is 3/8 of the full-aperture bandwidth, the focusing effect of the dominant wave is optimum. The panoramic data are processed by the proposed algorithm, and the refocused result is shown in Figure 10. Comparing Figure 6 and Figure 10, it can be found that the focus of the dominant wave is effective by the proposed algorithm.

## 4. Comparison and Quantitative Analysis of the Ocean Wave Refocusing Results

To sufficiently illustrate the effectiveness and superiority of the proposed algorithm, the refocusing results of the proposed algorithm are compared and quantitatively analyzed with the original SAR image, the refocusing SAR image based on half of azimuth phase speed [16,17,18,19,20,21,22] and the refocusing SAR image based on the optimum focus setting [25] in this section.

### 4.1. Comparison and Quantitative Analysis of the Ocean Wave Refocusing Results of Airborne L-Band SAR Data

In this subsection, the L-band data refocusing results of the proposed algorithm will be compared and quantitatively analyzed with the original SAR image, the refocusing SAR image based on half of azimuth phase speed and the refocusing SAR image based on the optimum focus setting.

#### 4.1.1. Comparison of the Ocean Wave Refocusing Results of Airborne L-Band SAR Data

The yellow rectangular area shown in Figure 5 is selected to compare. The comparison diagram is shown in Figure 11. Figure 11a–d correspond to the original SAR image of the L-band, the refocusing SAR image based on half of azimuth phase speed, the refocusing SAR image based on the optimum focus setting and the refocusing SAR image of the proposed algorithm, respectively.

By comparing Figure 11a–d, it can be found that the texture of the dominant wave is clear on the refocusing SAR image based on half of azimuth phase speed and the refocusing SAR image based on the optimum focus setting. After processing by the proposed algorithm, the texture of the dominant wave is clearer, and the details of the dominant wave are more abundant. Next, the quantitative analysis of the four images is carried out.

#### 4.1.2. Quantitative Analysis of the Ocean Wave Refocusing Results of Airborne L-Band SAR Data

To further illustrate the effectiveness and superiority of the proposed algorithm, a quantitative analysis of the dominant wave refocusing results will be carried out by calculating the contrast of the SAR image, relative normalized modulation of the image gray and modulation difference of positive and negative gray.

(1)Contrast of Images

The contrast of image is defined as
(17)$$C=\frac{\sqrt{\langle {\left[I^{2}\left(m,n\right)-\langle I^{2}\left(m,n\right)\rangle \right]}^{2}\rangle }}{\langle I^{2}\left(m,n\right)\rangle }$$
where I(m,n) is the amplitude of pixel (m,n) of the SAR image, and 〈⋅〉 is the average operator, which is used to calculate the average.

The larger the value of C is, the higher the image contrast is. In other words, the larger the value of C, the more obvious the ocean wave refocusing effect.

(2)Relative Normalized Modulation of Image Gray

The definition of relative normalized modulation of image gray is
(18)$$\left|\frac{\Delta I}{\bar{I}}\right|=\frac{1}{MN}\sum_{i=1}^{M}\sum_{j=1}^{N}\left|\frac{I\left(m,n\right)-\bar{I}}{\bar{I}}\right|$$

The relative normalized modulation of image gray $\left|\frac{\Delta I}{\bar{I}}\right|$ represents the ratio of the texture feature to the background feature, and it can be used to describe the information of the texture features. The larger the value of $\left|\frac{\Delta I}{\bar{I}}\right|$, the richer the information of the texture feature.

(3)Modulation Difference of Positive and Negative Gray

The definition of the modulation difference of positive and negative gray is
(19)$$S_{BD}=\Delta I_{B}-\Delta I_{D}$$
among them,
(20)$$\Delta I_{B}=\frac{1}{N_{B}}\sum_{j=1}^{N_{B}}\left(I_{j}^{B}-\bar{I}\right).$$
(21)$$\Delta I_{D}=\frac{1}{N_{D}}\sum_{j=1}^{N_{D}}\left(I_{j}^{D}-\bar{I}\right)$$
where $N_{B}$ and $N_{D}$ are the number of pixels in the positive and negative gray areas, respectively. $I_{j}^{B}$ and $I_{j}^{D}$ are the amplitude of each pixel in the positive and negative gray areas, respectively. The larger the value of $S_{BD}$, the greater the difference between the positive and negative gray.

The three parameters of the original SAR image, the refocusing SAR image based on the half of azimuth phase speed, the refocusing SAR image based on the optimum focus setting and the refocusing SAR image of the proposed algorithm corresponding to Figure 11 are shown in Table 3.

As shown in Table 3, the three parameters of the refocusing SAR image of the proposed algorithm corresponding to Figure 11d are more than doubled when compared with the original SAR image shown in Figure 11a, and they are increased by more than 10% compared with the refocusing SAR image based on half of azimuth phase speed shown in Figure 11b and the refocusing SAR image based on the optimum focus setting shown in Figure 11c. The results show that the proposed algorithm has the best wave focusing effect, followed by the refocusing SAR image based on the optimum focus setting.

### 4.2. Comparison and Quantitative Analysis of the Ocean Wave Refocusing Results of Airborne P-Band SAR Data

In this subsection, the P-band data refocusing results of the proposed algorithm will be compared and quantitatively analyzed with the original SAR image, the refocusing SAR image based on half of azimuth phase speed and the refocusing SAR image based on the optimum focus setting.

#### 4.2.1. Comparison of the Ocean Wave Refocusing Results of Airborne P-Band SAR Data

The yellow rectangular area shown in Figure 6 is selected to compare. The comparison diagram is shown in Figure 12. Figure 12a–d correspond to the original SAR image of the P-band, the refocusing SAR image based on half of azimuth phase speed, the refocusing SAR image based on the optimum focus setting and the refocusing SAR image of the proposed algorithm, respectively.

Comparing Figure 12a–d, it can be found that the ocean surface is uniform in the original image, and the texture characteristics of ocean waves are invisible. The texture of the dominant wave is clear on the refocusing SAR image based on half of azimuth phase speed and the refocusing SAR image based on the optimum focus setting. After processing by the proposed algorithm, the texture of the dominant wave is clearer, and the details of the dominant wave are more abundant. Next, the quantitative analysis of the four images is carried out.

#### 4.2.2. Quantitative Analysis of the Ocean Wave Refocusing Results of Airborne P-Band SAR Data

The three parameters of the original SAR image, the refocusing SAR image based on half of azimuth phase speed, the refocusing SAR image based on the optimum focus setting and the refocusing SAR image of the proposed algorithm corresponding to Figure 12 are shown in Table 4.

As shown in Table 4, the three parameters of the refocusing SAR image of the proposed algorithm corresponding to Figure 12d are more than 1.5 times compared with the original SAR image shown in Figure 12a, and that are increased by more than 30% compared with the refocusing SAR image based on half of azimuth phase speed shown in Figure 12b and the refocusing SAR image based on the optimum focus setting shown in Figure 12c. The results show that the proposed algorithm has the best wave focusing effect.

## 5. Discussion

### 5.1. Analysis of Algorithm Applicability

Aiming at the problem of ocean wave blur and the decorrelation caused by ocean wave motion in SAR integration time, a SAR imaging algorithm of ocean waves based on the optimum subaperture is proposed in this paper. The proposed algorithm is applied to the airborne field data, and it is compared with present algorithms, which proves the superiority of the proposed algorithm. Next, the applicability of the proposed algorithm will be analyzed.

(1) The longer the integration time, the more effective the proposed algorithm. The reason is that the longer integration time leads to a serious decorrelation of the ocean’s surface. In the case of serious decorrelation of the ocean’s surface, the ocean waves will suffer serious blurring, and the superiority of subaperture processing is more obvious. For example, for the P-band and L-band data applied in this paper, the integration time of P-band is much longer than that of L-band, and the processing results of P-band are improved more obviously.

(2) At present, the proposed algorithm is only applied to airborne SAR data. In fact, it can be well applied to other SAR data with long integration times, such as geosynchronous SAR data.

(3) The effect on SAR data with short integration times is not obvious. For example, for spaceborne SAR data such as ERS-2, the integration time is about 1 s, and the proposed algorithm is not feasible to significantly improve the quality of ocean waves.

(4) For SAR images almost completely covered by man-made targets such as ships and oil spills, it is impossible to select appropriate sub-block data according to Section 2.1. The proposed algorithm is not applicable in this case.

### 5.2. Prospects for Algorithm Applications

Clear SAR images are the basis for the acquisition of ocean wave information. Clear ocean wave images can be obtained using the proposed algorithm, and then, the wave spectrum can be analyzed to obtain the information about the ocean waves. The spectrums of the two pieces of data corresponding to the original SAR image and the SAR image processed by the proposed algorithm are shown in Figure 13. The results show that the spectrum characteristics of the waves processed by the proposed algorithm are more obvious, and the *PBR* calculated by Equation (8) is greatly improved, which is more conducive to the analysis and application of the subsequent wave spectrum.

## 6. Conclusions

Due to the random motion of waves, there is a problem of wave blurring when SAR observes the ocean, which seriously affects ocean information acquisitions such as the parameter retrieval of ocean waves and wind field inversion of the ocean’s surface. The blurring of SAR ocean wave images is mainly caused by the following two reasons. One is that the motion of the ocean waves leads to the blurring of the ocean waves. The other is that the decorrelation within the integration time seriously affects the imaging quality of the ocean waves. Aiming at the above two aspects, this paper proposed a SAR imaging algorithm of ocean waves based on the optimum subaperture. The proposed algorithm was applied to airborne SAR L-band and P-band data, and it was compared with present methods. The results show that the quality of the dominant wave of the L-band and P-band SAR data has been significantly improved by the proposed algorithm, which sufficiently demonstrates the effectiveness and superiority of the proposed algorithm.

In addition, the applicability of the proposed algorithm is analyzed. It is pointed out that the longer the SAR integration, the more obvious the effect of the proposed algorithm. The reason is that the longer integration time leads to serious decorrelation of the ocean surface. So, the superiority of subaperture processing is more obvious. Finally, the prospects for algorithm applications are given. It shows that the spectrum characteristics of the waves processed by the proposed algorithm are more obvious, which is more conducive to the analysis and application of subsequent wave spectrum.

## Figures and Tables

**Figure 1 sensors-22-01299-f001:**
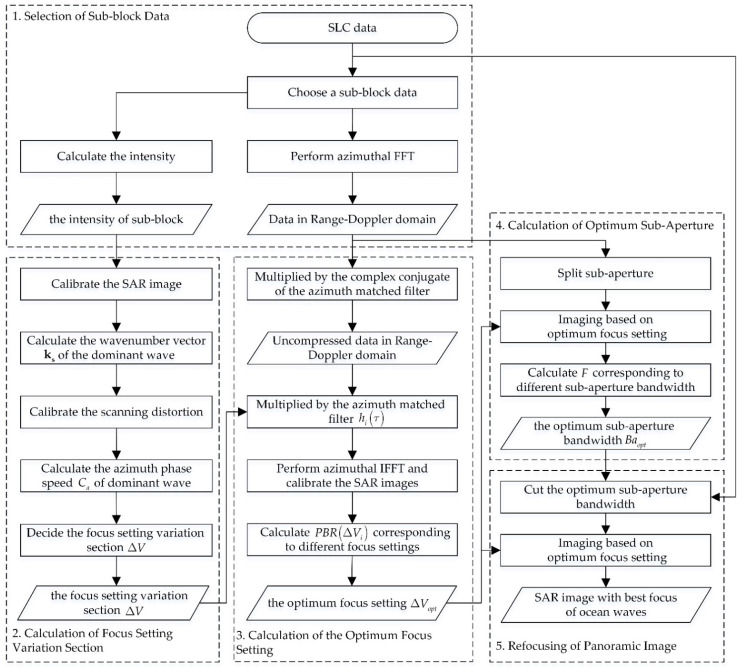
A flow chart of the proposed algorithm.

**Figure 2 sensors-22-01299-f002:**
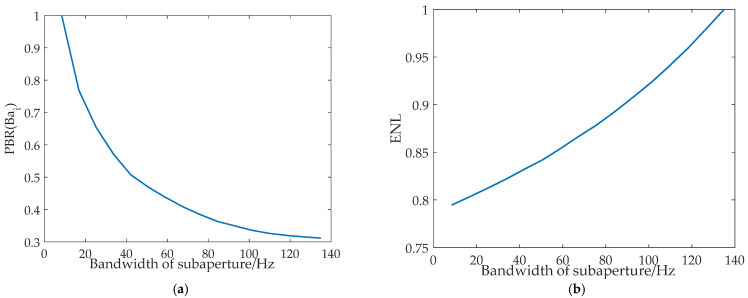
An effect analysis of subaperture bandwidth on ocean wave focusing. (**a**) $PBR\left(Ba_{i}\right)$ varies with the subaperture bandwidth; (**b**) ENL varies with the subaperture bandwidth.

**Figure 3 sensors-22-01299-f003:**
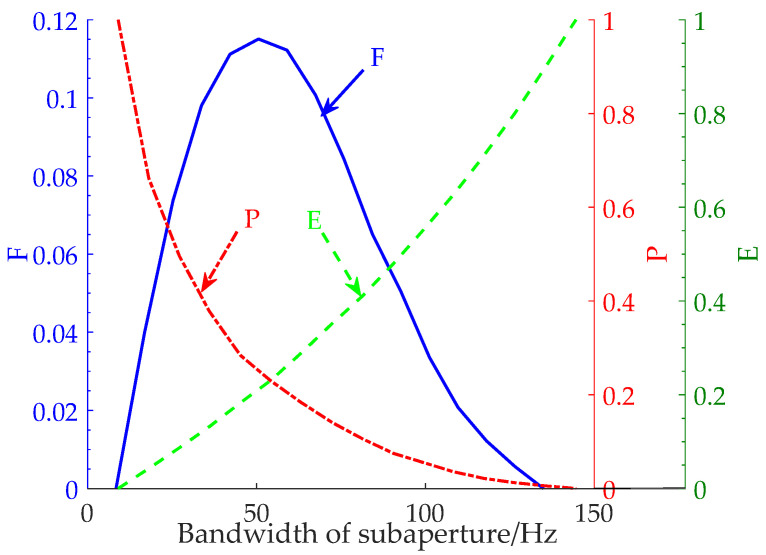
Curves of P,E,F vary with subaperture bandwidth. The red dotted line is the change in P with subaperture bandwidth. The green dashed line is the change in E with subaperture bandwidth. The blue solid line is the change in F with the subaperture bandwidth.

**Figure 4 sensors-22-01299-f004:**
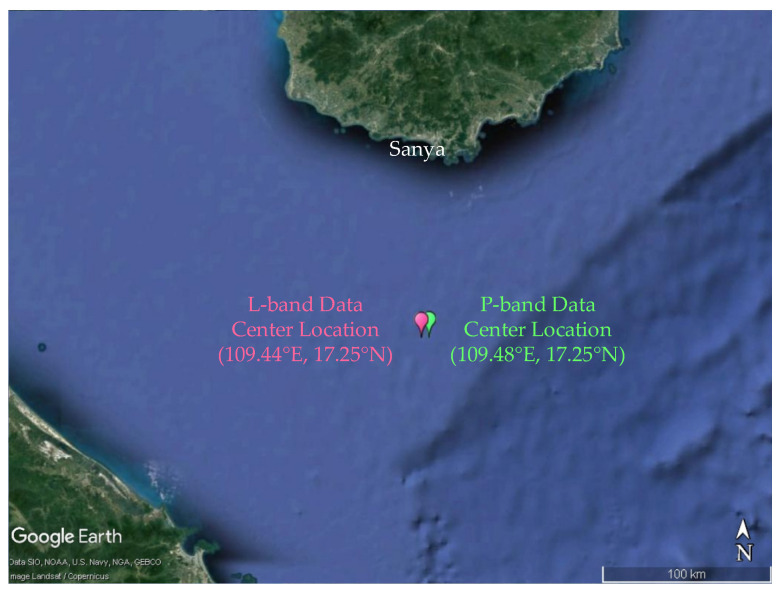
The geographic location of the SAR data on Google Earth. The location of the L-band data is marked in pink and that of the P-band is marked in green.

**Figure 5 sensors-22-01299-f005:**
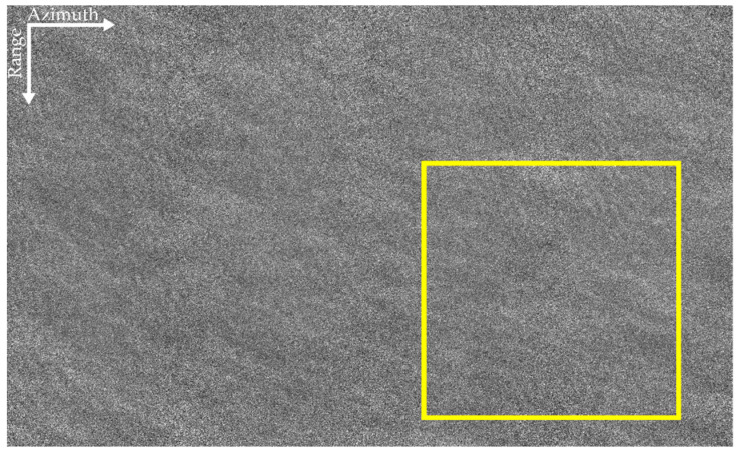
The L-band SAR image. The yellow rectangular area in the bottom-right corner shows the selected sub-block data.

**Figure 6 sensors-22-01299-f006:**
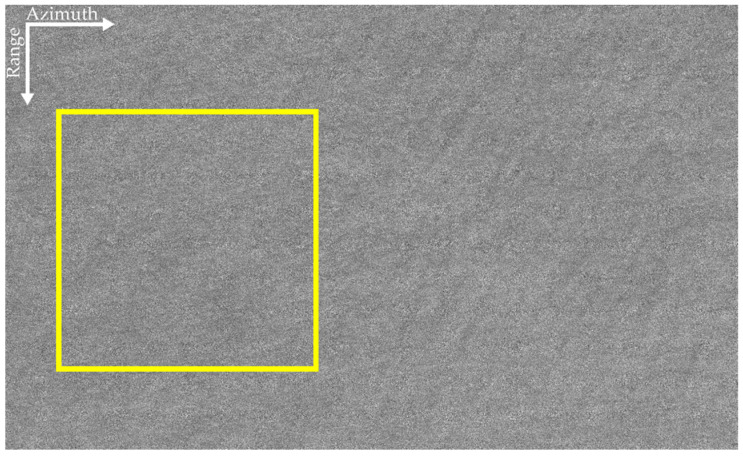
The P-band SAR image. The yellow rectangular area in the middle-left shows the selected sub-block data.

**Figure 7 sensors-22-01299-f007:**
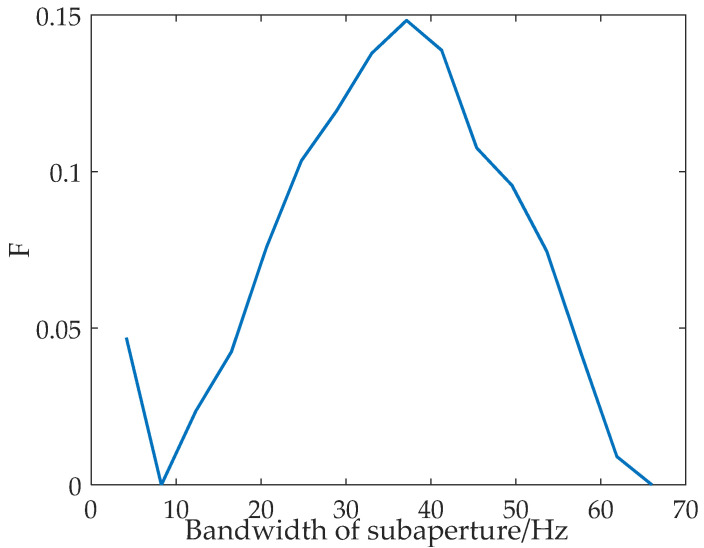
The curve of F varies with subaperture bandwidth.

**Figure 8 sensors-22-01299-f008:**
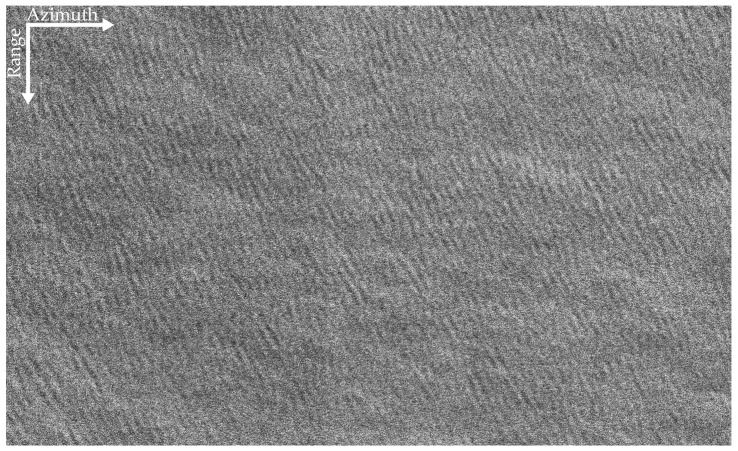
The L-band refocusing SAR image based on the proposed algorithm.

**Figure 9 sensors-22-01299-f009:**
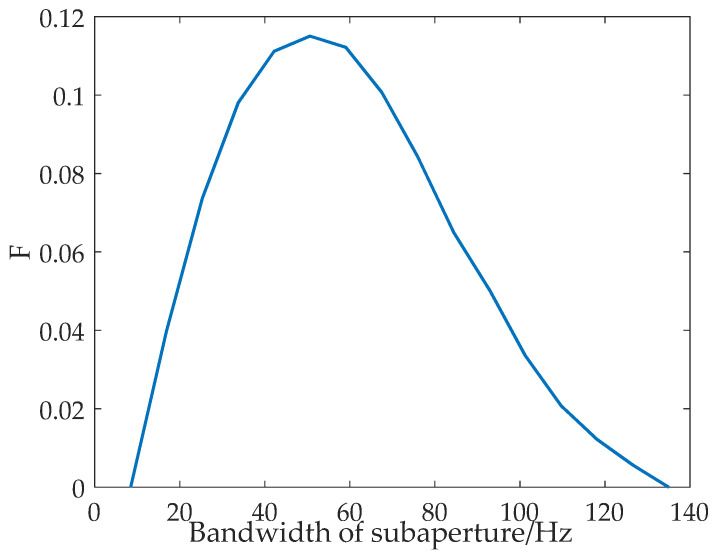
The curve of F varies with subaperture bandwidth.

**Figure 10 sensors-22-01299-f010:**
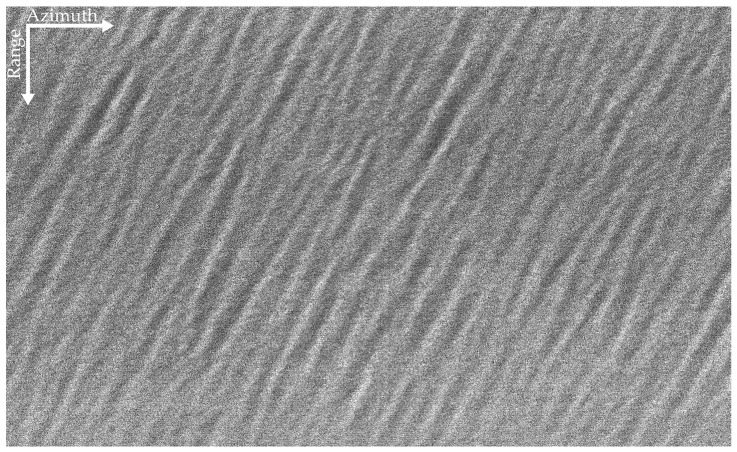
The P-band refocusing SAR image based on the proposed algorithm.

**Figure 11 sensors-22-01299-f011:**
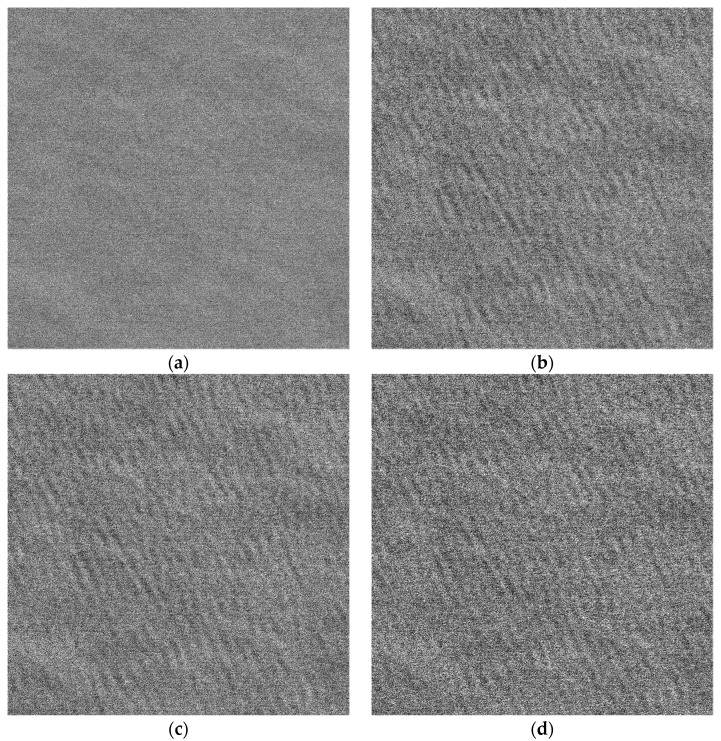
The comparison of rectangular area. (**a**) The L-band original SAR image; (**b**) the refocusing SAR image based on half of azimuth phase speed; (**c**) the refocusing SAR image based on the optimum focus setting; (**d**) the refocusing SAR image of the proposed algorithm.

**Figure 12 sensors-22-01299-f012:**
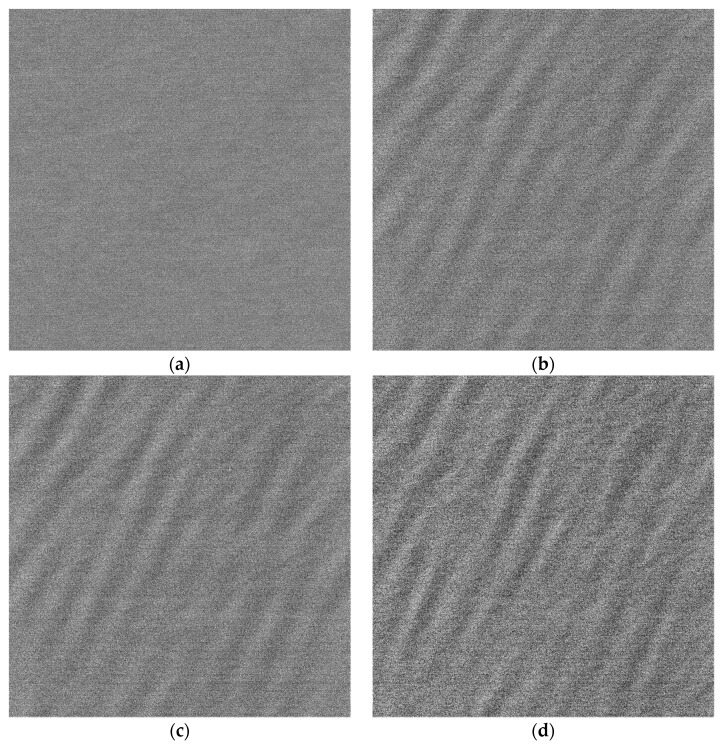
The comparison of rectangular area. (**a**) The P-band original SAR image; (**b**) the refocusing SAR image based on half of azimuth phase speed; (**c**) the refocusing SAR image based on the optimum focus setting; (**d**) the refocusing SAR image of the proposed algorithm.

**Figure 13 sensors-22-01299-f013:**
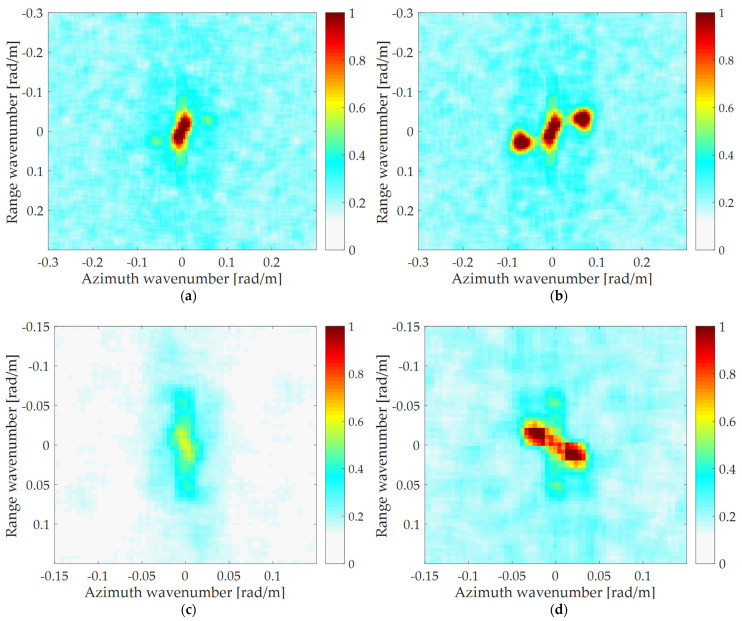
A spectrum comparison before and after processing by the proposed algorithm. (**a**) The spectrum corresponding to the L-band original SAR image (*PBR* = 2.41); (**b**) the spectrum corresponding to the L-band processed by the proposed algorithm (*PBR* = 7.34); (**c**) the spectrum corresponding to the P-band original SAR image (*PBR* = 2.68); (**d**) the spectrum corresponding to the P-band processed by the proposed algorithm (*PBR* = 7.78).

**Table 1 sensors-22-01299-t001:** The radar system parameters.

Parametric Name	Parametric Symbol	Parametric Value	
		L-Band	P-Band
Radar wavelength (m)	λ	0.23	0.5
Pulse length (us)	$T_{r}$	5.4	3
Radar bandwidth (MHz)	$B_{r}$	125	160
PRF (Hz)	PRF	900	1000
Platform speed (m/s)	V	132	134
Platform height (m)	H	8100	8600
Slant range of scene center (m)	$R_{0}$	13,000	11,300
Doppler bandwidth (Hz)	Ba	66	135
SAR integration time (s)	$T_{s}$	7	21

**Table 2 sensors-22-01299-t002:** The environment and wave parameters for the SAR data.

Data	13 September 2014	11 October 2014
Corresponding SAR data	L-band data	P-band data
Central location of SAR data	(109.44° E, 17.25° N)	(109.48° E, 17.25° N)
10 m wind speed (m/s)	3	6.5
Significant wave height (m)	0.5	1.5
Mean wave period (s)	5	8.5
Mean wave direction * (deg)	158	30

* Mean wave direction is the angle between the dominant wave propagation direction and the azimuth.

**Table 3 sensors-22-01299-t003:** The three parameters of the original SAR image, the refocusing SAR image based on half of azimuth phase speed, the refocusing SAR image based on the optimum focus setting and the refocusing SAR image of the proposed algorithm.

	Original Image	Result Based on Half of Azimuth Phase Speed	Result Based on Optimum Focus Setting	Result of the Proposed Algorithm
C	0.5791	1.0306	1.0434	1.1654
$\left|\frac{\Delta I}{\bar{I}}\right|$	0.2272	0.4075	0.4124	0.4726
$S_{BD}$	0.1427	0.2532	0.2562	0.2938

**Table 4 sensors-22-01299-t004:** The three parameters of the original SAR image, the refocusing SAR image based on half of azimuth phase speed, the refocusing SAR image based on the optimum focus setting and the refocusing SAR image of the proposed algorithm.

	Original Image	Result Based on Half of Azimuth Phase Speed	Result Based on Optimum Focus Setting	Result of the Proposed Algorithm
C	0.3412	0.6096	0.6279	0.8770
$\left|\frac{\Delta I}{\bar{I}}\right|$	0.1353	0.2443	0.2512	0.3526
$S_{BD}$	0.0847	0.1489	0.1533	0.2143

## Data Availability

Data sharing not applicable.

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
