# Peer review of "SAR Imaging Algorithm of Ocean Waves Based on Optimum Subaperture"

_sensors, 2022, doi:10.3390/s22031299_

Round 1
Reviewer 1 Report
Dear Editor and Authors,
The manuscript has been reviewed. In general, the study is a well presented paper. however, some changes are needed as follows.
- Study area is needed, such as description, map of SAR imagery and field data used in this study area.
- The flow chart should be simplified.
- Uncertainty analysis of the results are needed.
- The significance of this study should be highlighted in abstract and conclusion.
Author Response
We appreciate you very much for the constructive comments and suggestions. The response to the comments is attached.

Reviewer 2 Report
The manuscript, "SAR Imaging Algorithm of Ocean Waves Based on Optimum Subaperture, " proposes a SAR imaging algorithm of ocean waves based on optimum subaperture. The paper considers a worthy topic, and it is of interest to the remote sensing community. However, the manuscript is not well organized, and the structure is not strong. I strongly recommend revising the Journal's instructions for Authors to fit the arrangement of the document to the Journal's guidelines. For instance, there is no discussion section and, at least from my point of view, it should be a crucial part of the paper because of the new algorithm authors are proposing.
Here are my general comments:
First, I recommend revising and editing the introduction section because it is unclear why deriving the algorithm is essential and the basis for doing so. Lines 63-64 are not enough to provide enough justification or highlight the study's importance. Lines 66 to 81 looks at part of the material and methods section.
The flowchart showing the algorithm's structure is an excellent idea, but the quality of the figure is not good. Please consider enhancing the quality of figure 1.
About section 3, I think the proper name for this section should be "Applying the proposed algorithm". Validation implies a (or several) test (s) before. Also, it is risky to "validate" using only two images. Could the authors elaborate a little bit more on this?
How was the algorithm implemented? Through a code?
Section 4.1.1 is subjective since it is based on visual assessment. Please revise.
Why not include a discussion section? What are the caveats of the study? What's next? How did the authors compare their results to what is already available in the literature?
Please revise the conclusion section. The way that it is written is not correct. Conclusions should be concise and based on the findings.
Particular
Line 42. Please use a different word for "believe". Scientists do not "believe"; they conclude based on evidence.
Please enhance the quality of figures, especially those showing curves.
Author Response

(The authors gave the same response as above.)

Reviewer 3 Report
In this paper, an algorithm based on optimum subaperature is proposed to mitigate the effect of ocean waves on SAR imaging. To this end, the optimum focus setting of the ocean waves is calculated to refocus the dominant wave which help to obtain a clear SAR image. Experimental and analytical results on two datasets are carried out to demonstrate that the proposed algorithm is effective.
The paper is of practical value. I have the following comments, to be addressed before publication:
- Comparison aspect of the paper can be improved by testing other existing algorithms to demonstrate that the proposed method outperforms these methods. By doing so, the authors demonstrate that the algorithm offers a better result compared to the current methods.
- In the presented SAR images, comparison is left to visual inspection. I think it is more helpful to provide a quantitative metric for each figure on its caption to enable the reader to judge how well the performance for the algorithm is.
- Some other methods for SAR applications use noisy images but benefit from machine learning to cancel out the noise when performing a downstream task. I think some of these works should be included in the introduction. For example:
- Wang, Z., Du, L., Mao, J., Liu, B. and Yang, D., 2018. SAR target detection based on SSD with data augmentation and transfer learning. IEEE Geoscience and Remote Sensing Letters, 16(1), pp.150-154.
- Zhong, C., Mu, X., He, X., Wang, J. and Zhu, M., 2018. SAR target image classification based on transfer learning and model compression. IEEE Geoscience and Remote Sensing Letters, 16(3), pp.412-416.
- Lu, C. and Li, W., 2019. Ship classification in high-resolution SAR images via transfer learning with small training dataset. Sensors, 19(1), p.63.
- Rostami, M., Kolouri, S., Eaton, E. and Kim, K., 2019. Deep transfer learning for few-shot SAR image classification. Remote Sensing, 11(11), p.1374.
- Hamdi, I., Tounsi, Y., Benjelloun, M. and Nassim, A., 2021. Evaluation of the change in synthetic aperture radar imaging using transfer learning and residual network. Компьютерная оптика, 45(4), pp.600-607.
Although the above works are not on improving imaging but provide a different prospect on removing the noise effect using machine learning.
- It would be very helpful to show the effect of improving the SAR image quality on a downstream task to highlight practical benefit of the proposed algorithm.
Author Response

(The authors gave the same response as above.)

Round 2
Reviewer 2 Report
Many thanks to the authors for carefully addressing all my revisions. The manuscript has been substantially improved. It reads well, and it would make an excellent contribution to Sensors. Congratulations on a fine job. No further revisions are required.
Reviewer 3 Report
The authors have addressed my concerns.